# Comprehensive Two-Dimensional Gas Chromatography–Mass Spectrometry Analysis of Exhaled Breath Compounds after Whole Grain Diets

**DOI:** 10.3390/molecules26092667

**Published:** 2021-05-02

**Authors:** Kaisa Raninen, Ringa Nenonen, Elina Järvelä-Reijonen, Kaisa Poutanen, Hannu Mykkänen, Olavi Raatikainen

**Affiliations:** 1Institute of Public Health and Clinical Nutrition, University of Eastern Finland, P.O. Box 1627, FI-70211 Kuopio, Finland; ringa.nenonen@martat.fi (R.N.); elina.jarvela-reijonen@uef.fi (E.J.-R.); hannu.mykkanen@uef.fi (H.M.); olavi.raatikainen@uef.fi (O.R.); 2SIB Labs, University of Eastern Finland, P.O. Box 1627, FI-70211 Kuopio, Finland; 3VTT Technical Research Centre of Finland, P.O. Box 1000, FI-02044 VTT Espoo, Finland; kaisa.poutanen@vtt.fi

**Keywords:** exhaled breath, whole grain, rye, comprehensive two-dimensional gas chromatography–mass spectrometry, dietary fiber

## Abstract

Exhaled breath is a potential noninvasive matrix to give new information about metabolic effects of diets. In this pilot study, non-targeted analysis of exhaled breath volatile organic compounds (VOCs) was made by comprehensive two-dimensional gas chromatography–mass spectrometry (GCxGC-MS) to explore compounds relating to whole grain (WG) diets. Nine healthy subjects participated in the dietary intervention with parallel crossover design, consisting of two high-fiber diets containing whole grain rye bread (WGR) or whole grain wheat bread (WGW) and 1-week control diets with refined wheat bread (WW) before both diet periods. Large interindividual differences were detected in the VOC composition. About 260 VOCs were detected from exhaled breath samples, in which 40 of the compounds were present in more than half of the samples. Various derivatives of benzoic acid and phenolic compounds, as well as some furanones existed in exhaled breath samples only after the WG diets, making them interesting compounds to study further.

## 1. Introduction

Whole grain (WG) cereals are an important source of dietary fiber (DF) and micronutrients and are therefore acknowledged as part of the healthy diet in dietary recommendations [1,2]. Epidemiological studies and their meta-analyses have consistently shown high intake of WG to lower risk of chronic diseases and mortality [3,4,5], and associate negatively with obesity [6,7], type 2 diabetes [8,9,10], cardiovascular disease [11,12,13], and certain cancers [14,15]. However, the underlying physiological mechanisms are complex and unclear. Phenolic compounds in the fiber matrix of bran [16] are one proposed element for the protective effects of WG. Alkylresorcinols are present in the outer layers of wheat and rye grains and are known to be absorbed by humans. They have been detected in plasma and urine, and hence have been studied as a promising biomarker for WG wheat and rye in the diet [17].

People have variable metabolic responses to diets because of individual physiology and gut microflora [18]. Therefore, there has been an increasing interest in nutrigenomics, proteomics, and metabolomics to monitor metabolism from a wider perspective. Volatomic analysis of exhaled breath, used mainly for searching noninvasive biomarkers for diseases [19,20,21], could be used to characterize volatile organic compounds (VOCs) relating to various diets or specific foods, such as WG cereals. This research could lead to new information on the metabolic effects of WG foods and their association with health effects.

Exhaled breath is a potential noninvasive matrix to monitor metabolic changes induced by dietary modifications [22,23]. Foods contain numerous molecules which after digestion or metabolism by gut microflora are absorbed to the circulation. If they have a suitable boiling point, vapor pressure, and solubility, they can be excreted to exhaled breath. Currently, more than 3400 compounds have been identified in exhaled breath [24,25], with an average exhaled breath sample containing about 200 detected compounds [26]. However, there is wide interindividual variation since typically only a few dozen of the compounds are detected in every exhaled breath sample.

Although diet is known to cause variation in exhaled compounds [27], so far only a few pilot studies have been conducted to monitor the effects of diets on exhaled breath VOCs. Pioneering research in developing methods for exhaled breath analysis was done by Smith and Španěl who explored the effects of a meal [28] and glucose ingestion [29] to exhaled breath VOCs, as well as the effects of ketogenic diet on breath acetone levels [30]. Galassetti, Blake and colleagues studied exhaled breath compounds relating to diabetes [31,32] and monitored the effects of high-fat meals on exhaled breath VOCs [33]. They also studied exhaled breath VOC profiles relating to blood glucose [34,35,36] and lipid levels [37]. van Schooten et al. applied the breath analysis to monitor gastrointestinal diseases [38,39,40] and demonstrated a distinctive exhaled breath VOC profile after a gluten-free diet [41].

We have earlier demonstrated changes in exhaled breath VOC profiles by aspiration ion mobility spectrometry (AIMS) in diets differing in DF content (low-fiber diet vs. high-fiber) and type of bread (white wheat bread vs. sourdough fermented whole grain rye bread vs. white wheat bread enriched with modified rye fiber) [42,43]. However, the AIMS technology, regarded as a type of electronic nose, cannot identify the compounds responsible for the changes. Gas chromatography–mass spectrometry (GC-MS) is a standard technology for identifying VOCs [24,44] and multidimensional chromatography techniques, such as comprehensive two-dimensional GC-MS (GCxGC-MS), are utilized especially for characterization of compounds in metabolomic research, due to their increased separation capability [45]. However, they are not yet utilized for monitoring exhaled breath VOCs regarding to diets.

In this study, our aim was to pilot exhaled breath analysis with GCxGC-MS to explore VOCs relating to WG diets.

## 2. Results

About 260 VOCs were detected in 32 exhaled breath samples from 9 persons; of these VOCs, 40 were common, being present in more than half of the breath samples (Table 1). Carbon dioxide, isoprene, acetone, ethanol, 1-butanol, 2-propanol, benzene, benzaldehyde, methyl vinyl ketone, 2-butanone, phenol, hexanoic acid, and acetonitrile were found in all samples, but large individual differences existed in the other compounds. Additionally, 86 VOCs were tentatively identified by their MS spectra (Appendix A), whereas about 170 detected compounds remained unidentified.

Some derivatives of benzoic acid and phenolic compounds were detected in exhaled breath samples only after the WG diets (Table 2). Phthalic acid or phthalic anhydride (similarity index, SI 93 for both compounds) was found in 57% of the exhaled breath samples during the whole grain rye bread diet (WGR), in 11% of breath samples during the whole grain wheat bread diet (WGW), and in 6% of the background room air (BG) samples, but in none of the samples collected after the control diets containing refined wheat bread (WW). Benzoic acid was detected in 29% of breath samples during the WGR diet and in 11% of breath samples during the WGW diet, but in none of the exhaled breath samples during the WW diets and in 6% of BG samples. Furthermore, diphenyl ethanedione and benzamide were detected only after the WG diets, diphenyl ethanedione in 29% of breath samples during the WGR and benzamide in one participant during the WGR and the WGW.

Some furanones (ɣ-lactones) were also identified in the exhaled breath only during the WG diets: 5-dodecyldihydro-(3H)-furanone (in two participants after WGR), dihydro-4-hydroxy-2(3H)-furanone (in one participant after WGR and WGW) and dihydro-5-tetradecyl-2(3H)-furanone (in one participant after WGR and WGW).

We also detected several unidentified compounds having mass spectrum fragments 105, 77 and 51, which are typical for benzoic acid derivatives, and 107, 121, 135, and 149, typical for alkylphenols (potential degradation products of alkylresorcinols). However, none of the unidentified compounds were detected only in a particular diet period.

## 3. Discussion

We piloted exhaled breath analysis with GCxGC-MS to detect VOCs relating to WG diets. With this technology and the chosen method, about 260 compounds were detected from exhaled breath samples, and of these, 40 VOCs were present in more than half of the exhaled breath samples. Some benzoic acid and phenol derivatives, as well as furanone compounds, were detected more frequently after the whole grain diets.

GCxGC-MS-technology has better sensitivity and separation of compounds as compared to the traditional GC-MS, which make it suitable for non-targeted analysis of exhaled breath compounds. We have earlier analyzed exhaled breath VOCs by traditional GC-MS having the same column and same kind of sampling protocol [46], and about 40 VOCs were detected in the exhaled breath samples collected from healthy men. In the current GCxGC-MS protocol, the total number of detected compounds was approximately seven times more (about 260 compounds). The comprehensive GCxGC-MS technology is based on cryogenic modulator: effluent from the first column is trapped in the modulator for a given period (for 8 s in our method) before being released into the second column. This increases the sensitivity of the method remarkably as compared to traditional GC-MS.

GCxGC-MS technology also improved the separation of compounds as compared to traditional GC-MS. For example, we found two compounds giving almost identical mass spectra with isoprene, having only slightly different retention times. These are probably cis-1,3-pentadiene and trans-1,3-pentadiene which have been detected earlier in exhaled breath samples [47,48], or 1,4-pentadiene, which has been associated with smoking [49,50]. However, our participants were non-smokers. It is noteworthy that these compounds can be erroneously identified as isoprene, and therefore interfere the quantification of isoprene if they are not separated in the analysis. Exhaled breath isoprene has been studied extensively as a potential biomarker compound for cholesterol synthesis, though with controversial results [51,52]. It is possible that these compounds have interfered the quantification of isoprene in some studies.

Although the GCxGC-MS has advantages in sensitivity and selectivity, it also has drawbacks. Because of its sensitivity, the signal is easily overloaded when both the quantification and identification are challenged. This technology is suitable mainly for detecting compounds from challenging matrixes (having a multitude of compounds to be separated), but it is not very convenient for their quantification. The quantitative method should be optimized for each compound of interest separately, including calibration with breath mimicking conditions. Therefore, in this study we did not quantify the detected compounds. Exhaled breath VOCs might have multiple sources, and therefore it would be more relevant to monitor the changes in their levels rather than searching for specific biomarker compounds. However, nontargeted volatomic analysis can be used to select the relevant target compounds to monitor.

In total, 86 VOCs were tentatively identified from exhaled breath samples while about 150 VOCs remained unidentified, as their MS spectra were not found in the MS libraries. This indicates that there might still exist numerous unidentified molecules in the exhaled breath because GC-MS is a standard technology for identifying volatile compounds, and identification is mainly based on the MS libraries.

There is no analytical method available to monitor all the compounds in exhaled breath. For example, breath sampling method and thermal desorption (TD) adsorbents select the compounds, and GC column determine which compounds are chromatographically separated and can be detected. In this study, we chose the polar Nukol column for the first separative column because we have found it suitable for detecting endogenous gut-related exhaled breath VOCs [46], and non-polar Zebron ZB-35HT Inferno column for the second column due to its chemically different stationary phase compared to Nukol. By choosing other columns or TD adsorbent, different compounds could have been detected. It is noteworthy that most GC-MS analyses for exhaled breath VOCs are made by using general purpose nonpolar methylpolysiloxane columns containing 5% phenyl. With our protocol, we detected some common exhaled breath compounds such as isoprene, acetone, and ethanol, but we were unable to detect, for example, ammonia and methane (too small to detect with MS SCAN 35–300 *m*/*z*), or short-chain fatty acids (not enough sensitivity with MS SCAN mode [46]), although these compounds would be interesting in the perspective of nutrition and gut health [53,54,55]. We detected some compounds (Table 1; 3,4-dimethyl heptane, 3-pentanol, and methyl cyclopentane), which have not been reported in exhaled breath before. However, it should be pointed out that in our study, the identification was done only by the spectral library match, and was not confirmed with standard molecules (i.e., tentative identification) or retention indices. Mass spectra can be almost identical for some compounds, for example, for structural isomers (e.g., 2-methylbutane and n-pentane) or compounds with same structure with different length of alkyl chain (e.g., undecanal and tetradecanal). Therefore, the identification of VOCs in this study must be considered with caution.

Some benzoic acid and phenolic derivatives, as well as furanones, were detected from exhaled breath samples only after whole grain diets. It is possible that these VOCs are degradation products of phenolic compounds such as phenolic acids, alkylresorcinols and lignans from the DF complex in the bran. Phenolic compounds can be metabolized to various compounds by colonic fermentation and metabolism [56]; for example, benzoic acid can be formed from rye phenolics [57]. The compounds were detected mostly in the same exhaled breath samples, which indicate the same origin for these compounds.

Benzoic acid is known to be related to various foods but considered to have relatively low levels in the alveolar exhaled breath in the fasting state, since it is metabolized by liver and kidneys to hippurate within a few hours after oral dosing [58]. However, benzoic acid is formed also from whole grains in gut fermentation, which may explain the elevated levels in the fasting state in some individuals during WG diets. The exhaled breath samples were taken in the fasting state, but the fermentation rate may have been varied based on individual orocecal transit time and timing of eating WG. Estimation of the fermentation rate by breath hydrogen measurements [59] would be relevant when studying fermentation-related exhaled breath VOCs.

It´s noteworthy that the GC parameters used were not optimal for benzoic acid and phthalic acid/anhydride. Both compounds had wide tailing chromatograph peaks. This did not interfere with the identification of compounds, but may have affected sensitivity in their detection, and partly explains why these compounds were seen only in a minority of exhaled breath samples. Other GC parameters or technology should be used for analyzing these compounds more accurately.

Furanones are known to be formed in chemical reactions during charbroiling and seed oil cooking, and in Maillard reaction between sugars and amino acids [60,61]. They can also be metabolized from grain lignans such as matairesinol or 7-hydroxymatairesinol, or from enterolactone, a mammalian lignan, which is formed in the large intestine from plant lignans [62]. All these lignans have dihydro-2(3H)-furanone in their molecule structure. Enterolactone is considered a biomarker for high lignan intake in the diet [63], but high interindividual variation has been found in its absorption and metabolism [64]. To our knowledge, 5-dodecyldihydro-(3H)-furanone (CAS 730-46-1, also known as γ-palmitolactone), dihydro-4-hydroxy-2(3H)-furanone (CAS 5469-16-9, 3-hydroxy-γ-butyrolactone) or dihydro-5-tetradecyl-2(3H)-furanone (CAS 502-26-1, γ-stearolactone) have not been detected from exhaled breath before, unlike some other furanones [25,41]. Dihydro-5-tetradecyl-2(3H)-furanone has been detected from skin [25]. It would be interesting to monitor exhaled breath phenolic and furanone compounds and their levels in relation to different dietary sources, for example rye, using optimized analysis methodology for those compounds.

In this study, the randomized crossover protocol was used because the inter-individual variation of breath VOCs is known to be high [26]. In a crossover protocol, it is more likely that detected differences in breath VOCs are due to dietary changes because the other lifestyle factors are rather constant. Furthermore, most of the study participants were students or staff members of the Faculty of Health Sciences in the University of Eastern Finland and therefore likely to pay more attention to their eating than the average population in Finland. The participants consumed plenty of fruits and vegetables and received plenty of DF, and probably also phenolic compounds, also from sources other than the study breads. Therefore, the supply of DF remained higher than expected during the WW diets, being in the level of dietary recommendations. However, the total amount of consumed fruits and vegetables remained stable during the study, and there was a significant difference in the DF levels between WG and WW diets, as intended.

In conclusion, the GCxGC-MS technology, being sensitive and selective, offered some advantage for detecting exhaled breath VOCs. Benzoic acid derivatives, phenolic compounds, and furanones are potential compounds in monitoring metabolic effects of whole grains in exhaled breath. However, based on earlier reports by us [42,43] and others [22], it seems that it would be more relevant to monitor changes in the levels of multiple compounds or in VOC profiles rather than individual compounds when monitoring diet-related changes in exhaled breath VOCs.

## 4. Materials and Methods

### 4.1. Protocol

A randomized crossover manner dietary intervention was performed with 9 participants. They followed high-fiber diets containing either whole grain rye bread (WGR) or whole grain wheat bread (WGW) for 1 week in randomized order, and there were 1-week periods with refined (white) wheat bread (WW) before both test periods (Figure 1). At the end of the diet periods, exhaled breath samples in fasting state and parallel background air samples (room air samples, BG) were analyzed with GCxGC-MS technology. The RYEBREATH study was approved by the Ethics Committee of the Hospital District of Northern Savo (University of Eastern Finland, Hannu Mykkänen, 40/2015).

### 4.2. Study Participants

The participants were recruited into the RYEBREATH study with the campus advertisements in the University of Eastern Finland. They were healthy non-smoking Finnish men (2) and women (7) aged 21 to 59 years (average 31 years) and with BMI (body mass index) between 18.7 and 29 kg/m^2^ (average 23 kg/m^2^). The participants were advised to maintain their body weight and habitual lifestyle throughout the study, except the devised dietary modification for cereal content. All the participants provided written informed consent prior to participating in the study.

### 4.3. Diets

Participants followed three diets differing in consumed grain products. They were advised to consume 5–7 slices of white wheat bread per day during WW periods, 5–7 slices of whole grain rye bread during WGR period and 7–8 slices of whole grain wheat bread during WGW period. The commercial breads used in each period were: white toasts Vaasan Iso Paahto (DF 0.9 g/slice) and Oululainen Reilu Vehnä (DF 1.2 g/slice) during the WW periods, whole grain rye breads Fazer Real Ruis (DF 4.2 g/slice) and Porokylän leipomo PikkuKartano (DF 1.6 g/slice) during the WGR period, and wholegrain wheat breads Fazer Täysjyvä Paahto (DF 1.5 g/slice) and Vaasan Täysjyvä Isopaahto (DF 2.5 g/slice) during the WGW period. The study subjects were advised to avoid whole grain products during the WW periods and not to consume any rye except during the WGR period. Food items which typically increase gut fermentation and fermentative gases in the intestines, such as beans, cabbages, and xylitol products, were avoided throughout the intervention. A master´s student in clinical nutrition advised the participants weekly on the practical management of the diets. The participants filled in 4-day food records during each diet period and recorded the eaten amount of the test breads in a daily questionnaire. The food records were analyzed for nutrient intakes using the Diet32 software (version 1.4.6.3, Aivo Finland Oy, Turku, Finland).

The intakes of energy, protein, fat, and carbohydrates during the test diet periods were maintained at the same level during the intervention (Table 3). Only intake of DF was significantly different between the WW and WG periods. The participants consumed breads on average 159 g/day during the WW periods, 208 g/day during WGR, and 200 g/day during WGW, which covered 18% of energy intake in WW1, 19% in WW2, 24% in WGR, and 23% in WGW.

### 4.4. Exhaled Breath Analysis

End-tidal exhaled breath samples were taken with Bio-VOC^®^ samplers (Markes International Ltd., UK), which are made for capturing the last part of exhaled breath from the alveoli concentrated with VOCs excreted from the circulation (Figure 2a). Participants were trained to give an adequate sample. Before the sampling, participants brushed their teeth with toothpaste and rinsed the mouth effectively with water to stabilize the microbial fermentation in the mouth. They were sitting still without talking and breathing normally for a few minutes before sampling to standardize the ventilation. Then they gave a constant deep blow through the sampler. Exhaled breath samples were injected immediately after sampling into TD liners (fritted glass liner packed with Tenax GR, mesh 80–100, GL Sciences, Eindhoven, The Netherlands) using the Bio-VOC^®^ sampler as a gas syringe and adapter (self-made by sculpting from PTFE rod) to connect the sampler and the liner tightly (Figure 2b). The TD liner was closed with a storage cap (Brass Liner Blanking Cap, GL Sciences, Eindhoven, the Netherlands) and analyzed within 2–5 h (Figure 2c). The internal standard (1 µL 0.22 µg/µL acetone-d6 (Euriso-top, Saint-Aubin, France) in Milli-Q ultrapure water (Millipore, Bedford, MA, USA)) was injected to the TD liners 30–60 min before sampling by using gas tight syringe and the Bio-VOC^®^ sampler. The background room air samples were taken before breath samples by injecting the room air with the Bio-VOC^®^ sampler to the TD liners. The room air samples were otherwise handled in the same way as breath samples.

Analysis was performed with a GCxGC-MS device consisting of GCMS-QP2010 Ultra and AOC-5000 Plus injection system (Shimadzu Scientific Instruments, Columbia, MD, USA), Optic-4 multi-mode inlet (GL Science, Eindhoven, The Netherlands) and ZX-1 thermal modulator (Zoex Corporation, Houston, TX, USA) (Figure 2d). The injection was done with automated injection of AOC-5000 Plus to the inlet of Optic injector. The temperature of the inlet was at the beginning 35 °C for 2 min and then rose to the 200 °C at the rate of 18 °C/min. The injection was done in high-pressure mode with split 5 allowing the pressure of the inlet decrease temporarily during the injection. The injected sample was preconcentrated to the cryotrap after the injector at −100 °C for 7 min and released rapidly at 200 °C (temperature rise 60 °C/s) to the GC. VOCs were separated on two serial capillary columns; polar Nukol (0.25 μm thick phase, 0.25 mm internal diameter, 30 m long, Supelco, Bellefonte, PA, USA) and non-polar Zebron ZB-35HT Inferno (0.8 μm/0.18 mm/1 m, Phenomenex Torrance, CA, USA), separated by a cryogenic Zoex-modulator. The modulation was done with 8 s modulation time and 10–30% filling of the 5 L dewar of the liquid nitrogen. Carrier gas was Helium 4.6 (AGA, Espoo, Finland) with column pressure 150 kPa, column flow 2.14 mL/min, and linear velocity 45.5. The GC oven was programmed to be 35 °C for 10 min, then raised by 3 °C/min to 200 °C. The duration of the GC program was 70 min. The detection was done with MS SCAN 35–300 *m*/*z*, event time 0.02 s, and scan speed 20,000 unit/s. Temperature of the ion source was 200 °C and for MS interface 220 °C.

The data were analyzed using ChromSquare 2.2 data analysis software (Chromaleont, Messina, Italy). All the visible blobs in two-dimensional chromatograph were manually selected for identification. The tentative identification was performed by comparing their mass spectra with data from NIST 11 Mass Spectral library (The National Institute of Standards and Technology, Gaithersburg, MD, USA), Wiley Registry 10th Edition (John Wiley & Sons, Hoboken, NJ, USA), and Flavour & Fragrance Natural & Synthetic Compounds GCMS library FFNSC 2 (Shimadzu Corp., Kyoto, Japan). The identification was checked precisely for each blob by the researcher, but it was not confirmed with analytical standards or retention indices. Tentatively identified VOCs were reported only if they were found in more than a single exhaled breath sample. Four exhaled breath samples and five background room air samples were excluded because of technical problems in the GC-MS analysis. Siloxanes and polyethylene glycol compounds were excluded from analysis because they likely originate from column phases.

## 5. Conclusions

Exhaled breath VOCs reflect metabolism and lifestyle, thus having large interindividual variation and a lot of so far unidentified molecules. Since diet affects exhaled breath VOCs, they could be utilized in studying the metabolic effects of diets. The GCxGC-MS technology offers some advantage in making the detection of human VOCs sensitive and selective. Exhaled breath benzoic acid derivatives, phenolic compounds, and furanones are interesting compounds to study further when exploring the metabolic effects of whole grains.

## Figures and Tables

**Figure 1 molecules-26-02667-f001:**
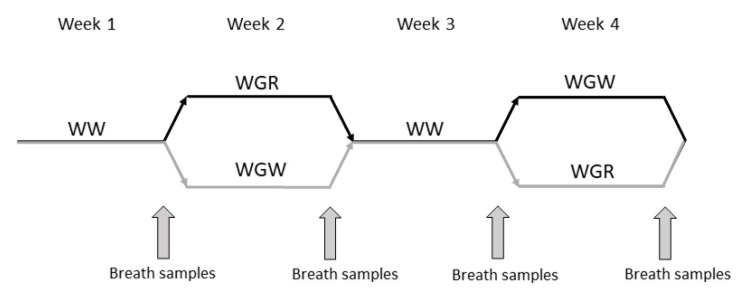
Study design of a randomized crossover manner dietary intervention with 9 participants. WW = control diet containing refined wheat bread; WGR = whole grain rye bread diet; WGW = whole grain wheat bread diet. Exhaled breath and parallel background room air samples were taken at the end of the diet periods.

**Figure 2 molecules-26-02667-f002:**
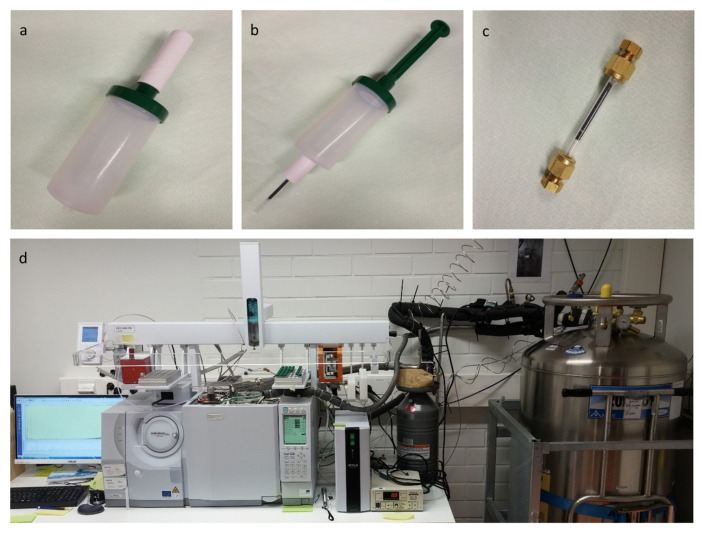
Exhaled breath analysis. End-tidal exhaled breath was sampled with the Bio-VOC^®^ sampler (**a**), injected into a glass liner containing Tenax GR absorbent (**b**), removed to laboratory in a sealed liner (**c**), and analyzed with comprehensive two-dimensional gas chromatography–mass spectrometry (**d**).

**Table 1 molecules-26-02667-t001:** Common ^1^ volatile compounds in the exhaled breath samples collected from the study participants after the diet periods and their presence in the background room air samples.

	Detected in % of Samples in			
Compounds	WGR	WGW	WW	BG
Carbon dioxide	100	100	100	100
Ethanol	100	100	100	100
Hexanoic acid	100	100	100	100
Acetophenone	100	100	100	100
1-Butanol	100	100	100	97
Benzene	100	100	100	97
Benzaldehyde	100	100	100	97
Methyl vinyl ketone	100	100	100	97
2-Butanone	100	100	100	97
Acetone	100	100	100	94
Phenol	100	100	100	94
2-Propanol	100	100	100	90
Acetonitrile	100	100	100	87
Isoprene	100	100	100	42
2,3-Butanedione	100	89	100	68
Toluene	100	33	81	74
3-Pentanol/2-Propanol, 2-methyl	86	100	100	97
Butanal	86	100	100	87
Hexanal	86	100	88	87
Heptanal	86	100	81	74
Octanal	86	89	94	23
Benzaldehyde, 2/4-methyl	86	89	75	90
Pentanal	86	89	69	68
n-Hexane	86	67	88	58
Nonanal	71	78	100	94
Acetaldehyde	71	56	81	77
3,4-Dimethyl heptane	71	33	75	35
D-Limonene	71	33	69	10
1,3-Pentadiene	71	33	69	0
Benzene, 1,4-dimethyl-	71	22	56	26
Dimethyl sulfide	57	78	81	0
Decanal	57	67	56	84
Methyl cyclopentane	57	56	63	42
6-Methyl-5-hepten-2-one	57	56	88	68
1-Propanol	57	44	81	39
Octane	57	44	56	17
Ethyl acetate	57	33	75	71
Styrene	57	33	69	29
p-Cymene	57	33	69	3
Heptane	43	56	75	71

^1^ present in >50% of the analyzed exhaled breath samples, WGR = whole grain rye bread diet (n = 7), WGW = whole grain wheat bread diet (n = 9), WW = refined wheat bread diet (n = 16), BG = background room air (n = 31).

**Table 2 molecules-26-02667-t002:** Volatile organic compounds detected only in the exhaled breath samples after WG diets.

			Detected in % of Samples in			
Compounds	RT	SI	WGR	WGW	WW	BG
Phthalic acid/Phthalic anhydride	40–69	93	57	11	0	6
Benzoic acid	30–69	94	29	11	0	6
Diphenyl ethanedione	61.7	92	29	0	0	0
5-Dodecyldihydro-(3H)-furanone	63.7	92	29	0	0	0
Benzamide	65.5	94	14	11	0	0
Dihydro-4-hydroxy-2(3H)-furanone	57.5	85	14	11	0	0
Dihydro-5-tetradecyl-2(3H)-furanone	67.7	89	14	11	0	0

RT = retention time (min), SI = similarity index, WGR = whole grain rye bread diet (n = 7), WGW = whole grain wheat bread diet (n = 9), WW = white wheat bread diet (n = 16), BG = background room air (n = 31).

**Table 3 molecules-26-02667-t003:** Mean daily intakes ^1^ of energy and nutrients during the 1-week diet periods (n = 9).

	WW1	WW2	WGR	WGW	*p*-Value ^2^
Energy, MJ	9.0 ± 1.7	9.3 ± 1.6	9.0 ± 1.8	9.0 ± 1.5	0.865
Carbohydrates, E%	42 ± 2	41 ± 4	42 ± 3	41 ± 4	0.706
Protein, E%	20 ± 3	20 ± 3	19 ± 2	20 ± 3	0.254
Fat, E%	35 ± 4	36 ± 5	35 ± 4	35 ± 7	0.954
Dietary fiber, g	24 ± 8	25 ± 8	36 ± 6 *	34 ± 1 *	<0.001

^1^ Values are means ± SD; ^2^ Statistical significance of the difference among the diet periods analyzed with Friedman’s test; * Different from WW periods, Wilcoxon´s test, *p* = 0.008; WW = diet with white wheat bread; WGR = whole grain rye bread diet; WGW = whole grain wheat bread diet; E% = percentage of total energy intake.

## Data Availability

The data presented in this study are available on request from the corresponding author. The original data are not publicly available due to privacy of participants.

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
