# Peer review of "Comprehensive Two-Dimensional Gas Chromatography–Mass Spectrometry Analysis of Exhaled Breath Compounds after Whole Grain Diets"

_molecules, 2021, doi:10.3390/molecules26092667_

Round 1

Reviewer 1 Report

The topic of the study might be of interest, but I have serious doubts about the strength of the conclusions that can be drawn from it. For one, the number of study participants (9) was inadequate for the search of biomarkers of a dietary intervention considering the huge number of factors that can affect the results. Second (and potentially even more important), the intervention itself was so minimal that any differences observed could easily be due to random factors. Case in point: one of the two refined white wheat breads contained as much as 1.2 g of DF per slice (the other contained 0.9 g), whereas one of the WGR breads contained 1.6 g DF/slice, and one of the WGW breads contained 1.5 g DF/slice. These differences are marginal at best, and I would be hugely surprised if they caused any measurable changes in the composition of exhaled breath.

The “identification” of the analytes is also inadequate. A comparison with library spectra on its own is not enough even for tentative identification. At the very least, it should be supported by retention indices. A cursory look at Table S1 quickly reveals that in many cases the identification must be wrong. For example, pentadecanal and hexadecanal are shown eluting before hexadecane, which is impossible. Even on a non-polar phase, the RI values of these two compounds are ~1700 and 1800, respectively. Considering that the z’ McReynolds constant for the Nukol stationary phase is 374, they should be eluting after n-C22 and 23, respectively.

In l. 31-33 the authors wrote “Epidemiological studies and their meta-analyses have consistently shown high intake of WG to lower risk of chronic diseases and mortality [3-5], and associate with obesity [6,7], type 2 diabetes [8-10], cardiovascular disease [11-13], and certain cancers”. As written, this statement suggests that high intake of WG promotes obesity, diabetes, etc., which is obviously not true. This sentence must be rephrased.

L. 48-49: Henry’s constant, which determines the partitioning of analytes between liquid and air, depends not just on “suitable boiling point and vapor pressure” (which both effectively mean the same), but also on the solubility of a compound (i.e. its polarity).

The agreed upon acronym for comprehensive two-dimensional gas chromatography is GC×GC, not “comprehensive 2D-GC”.

Reviewer 2 Report

The paper entitled “Comprehensive two-dimensional gas chromatography-mass spectrometry
analysis of exhaled breath compounds after whole grain diets” is focalized on a non-targeted
analysis of exhaled breath volatile organic compounds (VOCs) that was made by comprehensive
two-dimensional gas chromatography – mass spectrometry to explore compounds relating to
whole grain diets. Totally 113 VOCs were tentatively identified from exhaled breath samples;
among them various derivatives of benzoic acid and phenolic compounds, and some furanones
existed in exhaled breath samples only after the WG diets making them potential biomarker
compounds for whole grains.
In my opinion the paper could be accepted after minor revision. First, the paper could be
considered a pilot study, because only 9 subjects were investigated, and a lot of the compounds
detected are not still identified. This point should be emphasized in the text because the data,
even if interesting from a scientific point of view, are in a starting point and need to be supported
from a major number of samples.
Line 24: is not possible to define the differences identified in the study as potential biomarker,
please use a more suitable and weak term.

Reviewer 3 Report

The article " Comprehensive two-dimensional gas chromatography - mass spectrometry analysis of exhaled breath compounds after whole grain diets " is very interesting due to its applied nature.

I have few observations.

  1. 2D-GC-MS is an effective method for detecting volatile compounds. However, the volatile compounds identification was only achieved by comparison of NIST database is not enough. At least, the retention index should be added.
  2. Figure 1 is unnecessary.
  3. Table S1, any abbreviations, such as RT, SI, should be given or described below the table.
  4. Reference 1, 6 and 14 lack article number.

Round 2

Reviewer 1 Report

The authors addressed my comments appropriately